# Effects of a Twelve-Week Complementary Sports Program to Athletics Training on Motor Competence in Children Aged 6 to 10 Years Old—A Study Protocol

**DOI:** 10.3390/healthcare13172111

**Published:** 2025-08-25

**Authors:** Nataniel Lopes, Miguel Jacinto, Diogo Monteiro, Rui Matos, Sérgio J. Ibáñez

**Affiliations:** 1Facultad de Ciencia del Deporte, Universidade da Extremadura, 10071 Cáceres, Spain; nataniellopes@gmail.com (N.L.); sibanez@unex.es (S.J.I.); 2Research Center in Sports Sciences, Health Sciences and Human Development (CIDESD), 6201-001 Covilhã, Portugal; miguel.s.jacinto@ipleiria.pt (M.J.); diogo.monteiro@ipleiria.pt (D.M.); 3Life Quality Research Centre (CIEQV), 2040-413 Leiria, Portugal; 4ESECS—Polytechnic University of Leiria, 2411-901 Leiria, Portugal

**Keywords:** program, training, children, motor competence

## Abstract

Motor competence (MC) is defined as a global term that describes a person’s ability to be proficient in a wide range of motor acts. Based on this principle, we have created a training program that aims to determine the effect of 12 weeks of enriched athletics sports training with complementary motor activities on MC in children aged between 6 and 10 years old. The subjects will be divided into two groups: (i) the athletics training group (IG_A) that will participate in athletics training three times a week for 12 weeks, with 60 min sessions; and (ii) the athletics training + other activities group (IG_B) that will participate in athletics training twice a week and will have another activity training (gymnastics, handball, swimming, and motor games) for 12 weeks, with 60 min sessions. The two groups will be assessed at baseline and 12 weeks later. The KTK3+ will be used to assess MC. A between–within ANOVA-RM (2 [groups] × 2 [time points]) will be conducted. The results and conclusions of the implementation program will be presented in another study.

## 1. Introduction

Physical activity (PA) is widely recognized as a protective factor against a range of chronic diseases, including cardiovascular conditions, type 2 diabetes, and childhood obesity [1], and contributes to the short- and long-term improvement of the bone, muscle, and psychological health of children and adolescents [2]. Beyond its physiological benefits, PA plays a central role in the development of motor competence (MC) [1], an essential component of children’s physical literacy and a critical predictor of sustained engagement in an active lifestyle throughout life [3,4], and is related to performance in various sports [5].

MC can be defined as the ability to efficiently perform a broad repertoire of fundamental movement skills (FMSs), such as locomotor, stability, and object-control tasks [6,7]. The development of these skills is influenced by the dynamic interaction among individual (e.g., biological and hereditary), environmental (e.g., socio-cultural factors and prior experience), and task-specific factors (e.g., cognitive demand and mechanical properties) [6,8].

The evidence suggests that children with low levels of MC are more likely to show reduced PA participation, poorer physical fitness, and difficulties acquiring more complex movement skills [9,10,11]. These findings highlight the importance of early interventions, particularly during sensitive periods of motor development, when children are most responsive to motor learning through structured programs or educational strategies [12,13].

Structured interventions, such as guided motor games, quality physical education programs, and organized sports such as athletics, may be especially effective in promoting MC, as they provide diverse motor experiences and challenging FMSs. According to Lopes et al. [14], these improvements can positively affect overall physical fitness, which directly impacts cardiovascular health, body composition, and functional capacity. However, the literature lacks conclusive evidence regarding the specific effects of athletics-based programs on MC compared to multilateral or general physical education approaches [15].

Middle childhood, between the ages of 6 and 10, is a critical period for motor skill acquisition, characterized by rapid neuromuscular maturation, growth in physical capacities, and heightened adaptability to training stimuli. The evidence from school- and community-based interventions shows that structured PA (e.g., games, plyometric training, dance, and sports) or aquatics (e.g., swimming) consistently improves FMSs, coordination, and overall MC in this age group [16,17,18]. Aquatic programs have demonstrated significant gains in both global and specific motor skills, even over short durations, while land-based interventions have been shown to enhance complementary domains, such as strength, agility, and locomotor proficiency [19,20]. Collectively, the literature highlights that providing diverse, developmentally appropriate movement experiences during this sensitive growth window not only optimizes motor development but also supports long-term physical literacy and active lifestyle habits.

In this context, it becomes essential to assess the effects of systematic and structured training programs targeting MC development in early school-age children. Standardized tools, such as the Körperkoordinationstest für Kinder (KTK) [21], Test of Gross Motor Development of Ulrich [22], and Motor Assessment Batteries for Children (MABC) of Henderson and Sugden [23], have been widely used to evaluate MC due to their reliability and suitability for children aged 6 to 10 years.

Thus, the aim of this study is to design and evaluate a 12-week athletics-based training program for children aged 6 to 10, assessing its effects on MC. The hypotheses are as follows: (i) There will be significant improvement in MC after the 12-week intervention. (ii) There will be significant between-group differences in MC levels post-intervention. (iii) There will be significant improvement in MC of IG_B group after the 12-week intervention compared to the IG_A. (iv) The multi-activity program will lead to greater MC improvement in the IG_B group.

## 2. Materials and Methods

### 2.1. Study Design

This study follows a randomized, controlled experimental design with parallel groups (1:1 allocation ratio), aimed at evaluating and comparing the effects of two types of training protocols on MC in children aged 6 to 10 years. This methodology is used considering the conclusions of the systematic review and the fact that the main researcher is an athletics coach.

The participants will be ecologically assigned into one of two intervention groups, depending upon the training schedule chosen by their parents, according to their availabilities, and being given no prior indication of which of the two programs will function on which schedule: (i) Intervention Group A (IG_A) will receive athletics-based training exclusively, three times per week for 12 consecutive weeks, with each session lasting 60 min; and (ii) Intervention Group B (IG_B) will receive a combined program consisting of athletics-based training twice per week and one additional session per week involving a complementary activity (gymnastics, handball, or motor games), also over 12 weeks and with 60 min sessions.

All the participants will undergo MC assessments at two times: (i) at baseline (T0), prior to the beginning of the intervention; and (ii) at post-intervention (T1), immediately following the 12-week training period.

The post-intervention evaluations will take place three days after the final training session, always in the afternoon, to ensure consistency in the testing conditions. Figure 1 illustrates the participant flow and intervention timeline for the trial.

### 2.2. Participants

The study sample will be composed of children aged between 6 and 10 years, all of whom are to be active members of an athletics club located in the district of Leiria, Portugal. All the participants will be affiliated with the Portuguese Athletics Federation and covered by sports insurance, which includes protections regarding training sessions and data collection procedures. The selected age range fits into middle childhood, which is a critical period for motor development and the acquisition of fundamental motor skills. This age range aligns with previous research utilizing the KTK3+ battery, which has been validated and normed specifically for children within this developmental window [24,25], and was chosen because children between 6 and 10 years old are typically enrolled in primary school, where structured physical education and motor skill interventions are most effective at promoting motor competence and physical literacy [26,27]. To be eligible for inclusion in the study, the participants must meet the following criteria: (i) be aged between 6 and 10 years; (ii) be officially registered in the Portuguese Athletics Federation; and (iii) provide written informed consent signed by their legal guardians, along with the child’s verbal assent. The exclusion criteria are as follows: (i) any diagnosed physical or intellectual disability; (ii) any medical contraindications to physical exercise; (iii) failure to obtain parental consent; (iv) failure to complete the entire assessment protocol; (v) failure to complete a minimum of 80% attendance at training sessions in general and, in IG_B, failure to also complete a minimum of 80% attendance at complementary motor activities training sessions; and (vi) being outside the stipulated age range. The children meeting all the eligibility criteria will be randomly assigned to one of the two intervention groups, as described in the study design: (i) Intervention Group A (IG_A) or (ii) Intervention Group B (IG_B).

### 2.3. Ethical Approval

The current protocol has been reviewed and approved by the Scientific Ethics Committee of the Faculdad de Ciencias e del Deport de la Universidad da Extremadura, Spain (approval number: N°244/2024, 3 October 2024), and was developed following the Declaration of Helsinki for work with humans.

### 2.4. Instruments and Procedures

#### 2.4.1. Procedures

The intervention study was structured into several sequential phases to ensure methodological rigor and alignment with the study objectives: (i) SR—A comprehensive literature review was conducted to synthesize the current evidence on the effects of PA, organized sports, and structured training programs on MC in children; (ii) Program Design—the training protocol was developed based on the findings of the SR, incorporating best practices and evidence-based guidelines targeting fundamental motor skill development; (iii) Program Promotion—communication strategies will be employed to inform and engage the stakeholders, including the parents, coaches, and athletics club, about the program’s purpose and benefits; (iv) Definition of Procedures and Objectives—clear operational procedures and objectives for the intervention will be established to ensure fidelity in implementation and alignment with the research aims; (v) Participant Selection and Group Allocation—the sample will be defined, and the participants will be randomly assigned to either the athletics training group (IG_A) or the athletics training plus multi-activity group (IG_B); (vi) Baseline Assessment (T0)—an initial assessment will be conducted to establish the participants’ pre-intervention motor competence and anthropometric measures; (vii) Program Implementation—the 12-week training program will be administered to both intervention groups under controlled conditions; (viii) Post-Intervention Assessment (T1)—a final evaluation will be conducted 12 weeks later to assess the changes and outcomes related to the intervention; and (ix) Data Analysis and Interpretation—the collected data will be statistically analyzed to determine the effectiveness of the intervention, followed by a discussion and formulation of conclusions.

#### 2.4.2. Informed Consent

A comprehensive explanation of the study (materials and methods inclusive) by the research leader and the host institution will be provided to allow the participants/family members/tutors to be fully informed. The subject group will be given adequate time to decide about their participation. To do this, the participants, family members, and tutors must sign and deliver an informed consent form.

#### 2.4.3. Anthropometrics Assessment

These measurements will be taken by a researcher with extensive experience in anthropometric procedures. This will not only ensure greater intra-observer reliability, as these measurements will be taken by a single measurer, but will also eliminate the risk of inter-observer errors. The height will be measured using a stadiometer with a scale of 0.0 to 210 cm, while the children are barefoot and wearing only essential clothing (shorts and t-shirts). The Body weight will be measured using bioimpedance on a Tanita MC-780MAS Segmental digital scale (Tokyo, Japan). The body weight and height data will be used to calculate the Body Mass Index (BMI) using the formula BMI = Weight/(height)^2^.

#### 2.4.4. Motor Competence Assessment

Motor competence will be assessed using the KTK3+ test battery [28], which is an adaptation of the original Körperkoordinationstest für Kinder (KTK) by Kiphard and Schilling [21].

The original KTK primarily assesses gross motor coordination through balance and locomotor tasks; however, the KTK3+ introduces an additional eye–hand coordination (EHC) task, thereby expanding its capacity to evaluate manipulative skills, a core component of motor competence. This makes the KTK3+ a more holistic and ecologically valid tool, aligning with the theoretical frameworks that define MC as comprising locomotor, stability, and manipulative components [29,30].

The use of KTK3+ is therefore methodologically strategic, as it will allow us to comprehensively assess the multidimensional impact of both a mono-athletic sport and enriched athletic training program on children’s MC.

##### KTK3+ Test Application and Procedures

The test is made up of four tasks: Task 1, balancing backwards (BB); Task 2, jumping sideways (JS); Task 3, moving sideways (MS) [31]; and Task 4, eye-hand coordination (EHC). According to Gorla et al. [32], in the first task, dynamic balance is mainly checked; in the second, lower limb strength; in the third, laterality and spatial–temporal structuring; and in the fourth, locomotion, balance, and object control.

In the BB task, there are three trials per balance beam, which decrease in width as the test progresses (6.0 cm to 4.5 cm to 3.0 cm). The total number of steps is counted, with a maximum of 72 steps (or 8 steps in each trial per balance beam).

In the JS task, participants have to jump with two feet over a wooden slat for 15 s. The final score results from the sum of the number of jumps in both trials.

In the MS task, participants have to move sideways on a straight-line handling two wooden platforms for 20 s. The total score results from summing the number of times the participants put down a wooden platform and the number of times the participants step on the displaced wooden platform during both trials.

The EHC test is a valid and reliable product-oriented test [28] that determines the level of control of a tennis ball while conducting repetitive movements (i.e., left hand throw, right hand catch, followed by right hand throw, and left hand catch, etc.) as frequently as possible in a time-constrained task of 30 s [33]. The participants are free to use overhand and/or underhand techniques or a combination of both for throwing and catching. For this purpose, participants have to stand 1 m from a wall and throw the tennis ball at eye-level within a square (1 m^2^) taped on the wall, with the bottom side of the square 1 m above the ground. Participants perform this test twice, with the number of successful ball catches across both trials resulting in the test score.

The results of each item are compared with the normative values provided by the authors, and each item is assigned a quotient. The sum of the four quotients represents the global motor quotient (GMQ), which can be presented as a percentage or absolute value, making it possible to classify children for each age and sex according to their level of coordination development. For the total KTK3+ MQ-score, a classification of 5 levels of MC based on the normal distribution can be made [34]: (i) values below 70 are seen as an indicative of “severe gross MC disorder”; (ii) values between 71 and 85 are considered to represent “moderate gross MC disorder”; (iii) values between 86 and 115 are seen as “normal gross MC proficiency”; (iv) values between 116 and 130 are seen as “good gross MC proficiency”; and (v) values above 131 point to “high gross MC proficiency”. Before the initial assessment there will be practitioner training of the research team that will be assessing the KTK3+ to assure high inter-rater reliability.

### 2.5. Study Protocol

The protocol for both groups will be conducted according to the SR [35]. The subjects will be familiarized with the different exercises and tests during the preceding training period. The body composition and MC parameters will also be assessed at baseline (pretest) and post-intervention.

#### 2.5.1. Training Programs

The training programs will be implemented in both groups over 12 consecutive weeks, with 60 min training sessions three times a week, at Leiria’s Main Stadium, Dr. Magalhães Pessoa. The coaches will monitor the children’s attendance and absences at the training sessions, recording them on a specific attendance sheet.

##### Intervention Group A (IG_A) Training Program

Intervention Group A (IG_A) will participate in structured athletics training sessions three times per week, each lasting 60 min. The sessions will be held at the Municipal Stadium of Leiria—Dr. Magalhães Pessoa—on Tuesdays and Thursdays from 17:30 to 18:30, and on Saturdays from 09:00 to 10:00. The structure of each session will follow a consistent format. (i) Warm-up (5 min): Dynamic stretching or light jogging. (ii) Main phase (45 min): Focused athletics training targeting specific objectives—technique, speed, strength, and endurance. (iii) Cool-down (5 min): Static stretching or light jogging. The participants in this group will engage exclusively in athletics training, with no additional activities. The detailed 12-week training program is presented in Table 1.

Training Components OverviewRunning and technique Training

This component focuses on enhancing the biomechanics and efficiency of running. Exercises include drills with and without equipment (e.g., cones, hurdles, pins, and ropes) to promote better posture, stride mechanics, and coordination.

Speed Training

Speed sessions emphasize velocity over short distances (10–30 m), reaction starts, and agility circuits. Activities include varied start positions (sitting, lying), slalom runs, and hurdle sprints to improve explosive acceleration and coordination.

Strength Training

Strength development will be primarily based on bodyweight resistance exercises (e.g., planks, squats, and jumping jacks) and light resistance tools (e.g., 1–2 kg medicine balls, ropes, and kettlebells). Training will follow a circuit-based format using Tabata intervals (20 s work, 10 s rest). Each session will include 4–5 rounds of 4 exercises, with 2 min rests between rounds, focusing on proper technique, functional strength, and muscular endurance.

Resistance Training

Endurance activities will be integrated through continuous running (12–15 min) and structured motor games (e.g., relays, object transport tasks, and team challenges). Given children’s limited motivation for isolated endurance drills, the activities are designed to be engaging and enjoyable, promoting a long-term affinity for aerobic exercise.

##### Intervention Group B (IG_B) Training Program

Intervention Group B (IG_B) will be engaged in three weekly training sessions, each lasting 60 min. The sessions will take place at the Municipal Stadium of Leiria—Dr. Magalhães Pessoa—on Tuesdays and Thursdays from 18:30 to 19:30, and on Saturdays from 10:00 to 11:00. This group will follow a blended training approach: (i) two sessions per week (Tuesdays and Saturdays) focused on athletics, replicating the exact training content as IG_A; (ii) one weekly session (Thursdays) dedicated to complementary motor activities, alternating between gymnastics, handball, swimming, and motor games. The complementary session is strategically scheduled for Thursdays, a day with normally higher attendance and positioned between the two athletics sessions, to ensure optimal participation and recovery. A detailed breakdown of the 12-week program is provided in Table 2.


*Training Objectives*


The IG_B program is designed to evaluate the impact of combined athletics and multi-sport exposure on motor competence development. This group will receive the same foundational athletics training as IG_A, but with an added component of diversified motor experiences. The goal is to determine whether this combined training model will provide greater MC benefits than an athletics-only approach.


*Complementary Activities*

*Gymnastics and handball*


These sessions will emphasize foundational movement skills (FMSs) rather than technical sport-specific mastery. The intent is to expose participants to a variety of motor patterns through basic posture, balance, coordination, object manipulation, jumping, and rolling tasks. In handball, an emphasis will be placed on ball handling, passing, receiving, and spatial awareness, helping to enrich motor diversity.


*Swimming*


In week 10, swimming will be introduced to develop aquatic motor skills, including propulsion and object manipulation in water. This aims to broaden the scope of physical literacy and neuromuscular adaptation through exposure to aquatic environments.


*Motor games*


Motor games will be employed to stimulate key motor competencies, such as locomotion, coordination, balance, and object manipulation. These games are intentionally chosen for their educational value and children’s familiarity with them, ensuring engagement and effective skill acquisition. Activities will include cooperative games, relay formats, object transport tasks, and dynamic obstacle circuits.

## 3. Outcomes

The assessments will be conducted at the Main Stadium of Leiria, Dr. Magalhães Pessoa. We will use a gym prepared for gymnastics training. It is a large, well-equipped space that provides an atmosphere of concentration and comfort during testing. The MC assessment will occur on a Tuesday (i.e., three days after the last session—Saturday). In addition, assessments will take place during the training session period, meaning that IG_A will be tested at 17h30 and IG_B at 18h30. Given the spatial, human, and material conditions, it is possible for children from the same group to take the tests simultaneously, facilitating the process. Before carrying out the tests, the research group will provide the children with as much information as possible to avoid any doubts and answer all their questions clearly.

## 4. Statistical Analysis

The required sample size was calculated using the G*Power software (3.1.9.4; Heinrich Heine University Düsseldorf, DE). As the aim is to detect differences in the KTK3+ between the two groups, and considering that the analysis will be performed on the outcomes, a between–within ANOVA-RM (2 [groups] × 2 [time points]) will be conducted, anticipating a ‘large’ effect size (f = 0.4) based on similar previous studies using the KTK3+ [26,27,36], with an α = 0.05, a statistical power of (1 − β) = 0.95, correlated dependent variables with an r = 0.50, and a violation of sphericity (ε) = 0.80. This analysis will require a total sample size of 18 individuals per group. Thus, the study should start with an additional 15% of the planned sample size for each group to account for potential dropouts as suggested by several authors (e.g., [37]). It will also be comparable to previous school-based MC studies that employed the KTK3+ as an outcome measure in early childhood and primary education contexts [24,25], considered adequate for the study objectives and the planned statistical analyses. The means and standard deviation will be calculated for all the studied variables. The normality and homoscedasticity will be verified with the Shapiro–Wilk (n < 50) and Levene’s tests, respectively. The statistical analyses will be conducted in IBM SPSS Statistics version 27. The referred to ANOVA-RM will be applied to each of the four KTK3+ battery tests independently and to the KTK3+ battery results as a whole, namely to its raw results and to its Global Motor Quotient. Thus, we will be conducting six distinct analyses and will extract conclusions from each of them, with the latter two being the most important, as global results. Nevertheless, if specificities can be found for the other four, we intend to address and to consider them.

## 5. Discussion

The present study sought to examine and compare the effects of two distinct training program interventions on MC in children aged 6 to 10 years, a developmental period recognized as crucial for the acquisition of FMSs and long-term physical literacy [38,39]. As MC has been widely recognized as a key factor in promoting health and well-being and has also been linked to physical health and psychosocial aspects throughout childhood [3], children with higher motor proficiency are more likely to be physically active, creating a positive feedback loop in which motor development facilitates physical activity, and physical activity, in turn, enhances motor skills. Children who develop strong motor skills often report higher levels of self-efficacy, which encourages regular participation in physical and sports activities, contributing to the adoption of healthy lifestyle habits [38].

Grounded in the findings of a preliminary SR [35], we designed two intervention protocols, one based on athletics alone (IG_A) and the other using a complementary sport program to athletics training with a multi-activity structure (IG_B), both aiming to enhance the core components of MC: locomotion, object manipulation, and postural stability [6].

As the main researcher is an athletics coach, our aim with this study was to propose this training methodology for athletics, aimed at the 6–10 age group, to see whether it is feasible to promote the acquisition of sport-specific skills in athletics without compromising the development of MC. It is important to note that the intervention program that will be implemented essentially aims to show that it is possible to achieve better CM results with just one change to one of the weekly training sessions. In addition, it is also important to say that if there are results in this regard, athletics clubs will be challenged to incorporate these complementary activities into their programs.


*Multi-component Design and Theoretical Foundations*


The IG_B program, which integrates athletics, gymnastics, handball, motor games, and swimming, was intentionally structured to reflect the principles of motor development and neuroplastic adaptation. Theoretical models, such as the Developmental Model of Motor Competence [39] and the Mountain of Motor Development [40], emphasize that MC evolves through varied and context-rich physical experiences. This model informed the IG_B design, aiming to expose children to a diverse range of motor tasks.

The principle of variability of practice [41] supports this design, suggesting that children benefit from being exposed to a variety of tasks and contexts, as this enhances skill transfer and neural adaptation. Studies by Kriemler et al. [42] and Cohen et al. [43] confirm that diverse, school-based physical activity programs lead to greater improvements in motor competence compared to narrow-scope interventions.

The empirical findings support this approach: multi-component interventions, particularly those combining structured sport-specific activities with play-based components, are associated with improvements in FMSs compared to traditional PE or single-sport programs [30,44]. For instance, gymnastics promotes postural control and dynamic balance, handball fosters object manipulation and spatial awareness, and swimming enhances coordination and bilateral movement patterns [11,45]. These varied experiences contribute to the generalizability and transfer of motor skills to new physical contexts.

*Critical Comparison: IG_A* vs. *IG_B*

While the IG_A program focuses exclusively on athletics, allowing for repeated practice and a potentially deeper refinement of specific movement patterns (e.g., sprinting, hurdling), it lacks opportunities for manipulative and aquatic skill development. Although some research supports early specialization in contexts such as talent identification [46], the consensus in the developmental literature suggests that early diversification yields greater long-term benefits in MC, enjoyment, and sustained PA engagement [13,47].

Moreover, the inclusion of motor games and cooperative tasks in the IG_B program was intended not only to enhance physical skills but also to promote cognitive and socio-emotional competencies, consistent with the Embodied Cognition Framework [48]. The alignment of physical and cognitive engagement is particularly relevant for this age group, in which executive functions co-develop with movement proficiency [49].

We believe that this study can help us understand the effects of an enriched athletics training program on the development of MC. If the program meets our expectations, we can help fill in some of the gaps that exist in the approach to younger people.


*Methodological Considerations and Limitations*


The lack of sport-specific studies identified in the SR limits the ability to compare our future findings with the existing literature. The field remains dominated by generalized extracurricular and PE-based interventions [42], highlighting a gap in the understanding of how structured sport disciplines uniquely contribute to MC development.

Another limitation relates to the duration and intensity of the intervention. While a 12-week period with tri-weekly sessions aligns with prior interventions (e.g., Cohen et al. [43]), long-term follow-up assessments will be essential to assess skill retention and behavioral transfer to out-of-school PA contexts.


*Contributions and Future Directions*


This study offers an innovative contribution by operationalizing a multi-component training program based on the SR [35], with potential for curricular and extracurricular applications, essentially aiming to show that it is possible, with just one modification to one of the weekly training sessions, to achieve better results in MC. If the hypothesis regarding the IG_B program’s superiority is confirmed, the findings would support policy and curriculum recommendations that promote multi-sport exposure in early childhood, thus counteracting early specialization trends often observed in youth sports [50].

We believe this pilot study contributes meaningfully by proposing a concrete and replicable framework that may help researchers and practitioners implement evidence-informed physical education and training. If the hypothesis that the enriched athletics program (IG_B) will lead to greater MC improvement is confirmed, it will reinforce existing evidence that early diversification, rather than early specialization, is more effective in promoting long-term physical development and engagement in physical activity [47]. If so, and as previously said, efforts will be made to encourage athletics clubs to incorporate these complementary activities into their programs.

Future research should explore the longitudinal effects, stratify the samples by the initial MC levels, and incorporate the psychosocial outcomes (e.g., perceived competence, motivation) which are known mediators of sustained engagement in PA [11,51]. Moreover, randomized controlled trials (RCTs) with matched-pair designs could provide more rigorous evidence on the causal mechanisms linking multi-activity training to motor and cognitive outcomes.

## 6. Conclusions

The intervention based on the present protocol proposes an enriched athletics sports training program, adding complementary activities to athletics to improve MC in children aged between 6 and 10 years old. By addressing locomotor, manipulative, and stability components through diversified and developmentally appropriate activities, it is expected that this program will demonstrate the potential to foster broad-based motor development.

In short, we believe that combining sport-specific training with playful, skill-oriented games and activities can represent a more effective and engaging model for childhood physical activity interventions than traditional single-sport formats.

## Figures and Tables

**Figure 1 healthcare-13-02111-f001:**
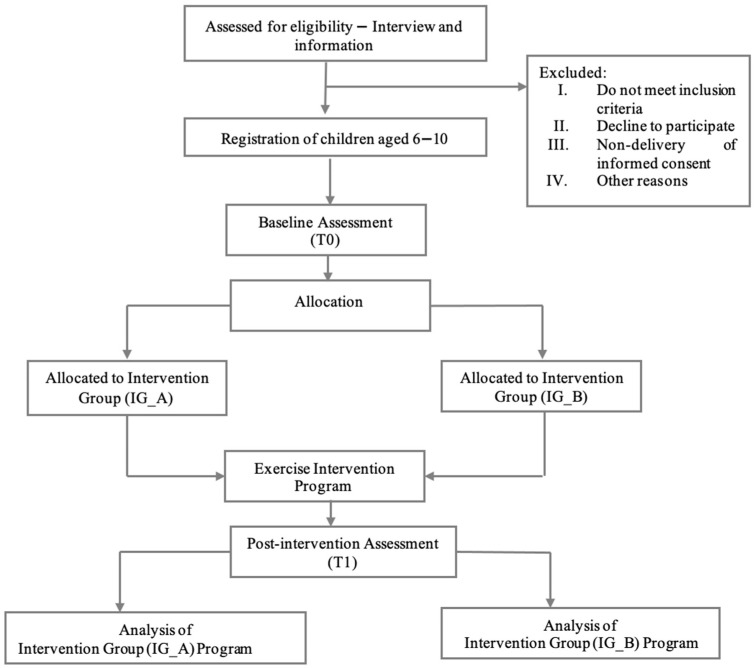
Timeline of the study design.

**Table 1 healthcare-13-02111-t001:** Intervention Group A (IG_A) program.

Week	Sports	Training Objective	Exercises Prescription	Volume
1	Athletics	Technics and Speed	Main part: Running technics (Skipping’s). Speed: Various starts—sitting, lying down, etc.	60 min
	Athletics	Speed and Strength	Main part: Agility and coordination circuit and general physical condition circuit.	60 min
	Athletics	Strength and Resistance	Main part: General physical condition circuit and varied games to stimulate resistance.	60 min
2	Athletics	Technics and Speed	Main part: Hurdles training technic and running at high speed with 4, 5, and 6 low hurdles.	60 min
	Athletics	Speed and Strength	Main part: Agility and coordination circuit and multi-throw with medicine ball (1–2 kg).	60 min
	Athletics	Strength and Resistance	Main part: General physical condition circuit and aerobic running (10–15 min).	60 min
3	Athletics	Technics and Speed	Main part: Running technique training with cones and pins. Speed: Speed games.	60 min
	Athletics	Speed and Strength	Main part: Fast running over short distances and multi-jumps in a sandpit.	60 min
	Athletics	Strength and Resistance	Main part: Strength training on stairs and resistance games (example: Formula 1).	60 min
4	Athletics	Technics and Speed	Main part: Race modeling (pins at different distances). Speed: Short-duration speed.	60 min
	Athletics	Speed and Strength	Main part: Pursuit race and hurdles races and general physical condition circuit.	60 min
	Athletics	Strength and Resistance	Main part: General physical condition circuit and resistance games (example: suicide).	60 min
5	Athletics	Technics and Speed	Main part: Long jump technics training. Speed: Short-duration speed (15 to 20 m).	60 min
	Athletics	Speed and Strength	Main part: Block star training and strength training on hills (fast running on short hills).	60 min
	Athletics	Strength and Endurance	Main part: General physical condition circuit and aerobic running (10–15 min).	60 min
6	Athletics	Technics and Speed	Main part: Block start training and race modeling. Speed: Various starts—sitting, lying, etc.	60 min
	Athletics	Speed and Strength	Main part: Agility and coordination circuit and general physical condition circuit.	60 min
	Athletics	Strength and Endurance	Main part: Agility and coordination circuit and long jump and resistance games (relay).	60 min
7	Athletics	Technics and Speed	Main part: Relay training technics and circuit for agility and coordination (slalom).	60 min
	Athletics	Speed and Strength	Main part: Various starts (sitting, lying down) and strength training on stairs.	60 min
	Athletics	Strength and Endurance	Main part: Strength training using only body weight and aerobic running (10–15 min).	60 min
8	Athletics	Technics and Speed	Main part: High jump training. Speed: Short-duration speed (15 to 20 m).	60 min
	Athletics	Speed and Strength	Main part: Race modeling (pins at 3 or 4 m distance). Strength training: Body weight.	60 min
	Athletics	Strength and Endurance	Main part: General physical condition circuit and resistance games (ex.: relay 4 × 400 m).	60 min
9	Athletics	Technics and Speed	Main part: Running technics (Skipping’s). Speed: Speed games (ex.: Divisions game).	60 min
	Athletics	Speed and Strength	Main part: Short-duration speed (15–20 m) and strength training with medicine balls.	60 min
	Athletics	Strength and Endurance	Main part: Multi-jumps in sandpit and aerobic running (10–15 min).	60 min
10	Athletics	Technics and Speed	Main part: Running technique training with cones and agility and coordination circuit.	60 min
	Athletics	Speed and Strength	Main part: Speed games (slalom and relay 4 × 40 m). Multi-throwing with medicine balls.	60 min
	Athletics	Strength and Endurance	Main part: Strength training using only body weight and aerobic running (15–20 min).	60 min
11	Athletics	Technics and Speed	Main part: High jump training. Speed: Short-duration speed (30 to 40 m).	60 min
	Athletics	Speed and Strength	Main part: Speed (relay 4 × 40 m). Strength: Multi-jumps (boxes, ropes, stair, hurdles, etc.).	60 min
	Athletics	Strength and Endurance	Main part: General physical condition circuit and resistance games (ex.: Formula 1).	60 min
12	Athletics	Technics and Speed	Main part: Hurdles training technic and running at high speed with 4, 5, and 6 low hurdles.	60 min
	Athletics	Speed and Strength	Main part: Agility and coordination circuit and general physical condition circuit.	60 min
	Athletics	Strength and Endurance	Main part: Strength training using only body weight and aerobic running (15–20 min).	60 min

**Table 2 healthcare-13-02111-t002:** Intervention Group B (IG_B) program.

Week	Sports	Training Objective	Exercises Prescription	Volume
1	Athletics	Technics and Speed	Main part: Running technics (Skipping’s). Speed: Various starts—sitting, lying down, etc.	60 min
	Gymnastics	Technics and Initiation	Main part: Basic fundamentals of gymnastics (learning basic posture exercises).	60 min
	Athletics	Strength and Resistance	Main part: General physical condition circuit and varied games to stimulate resistance.	60 min
2	Athletics	Technics and Speed	Main part: Hurdles training technic and running at high speed with 4, 5, and 6 low hurdles.	60 min
	Handball	Technics and Initiation	Main Part: Relationship with the ball and fundamentals (passing, receiving, and shooting).	60 min
	Athletics	Strength and Resistance	Main part: General physical condition circuit and aerobic running (10–15 min).	60 min
3	Athletics	Technics and Speed	Main part: Running technique training with cones and pins. Speed: Speed games.	60 min
	M. Games	Technics and Initiation	Main Part: Games to develop general MC (rhythm and cooperation games).	60 min
	Athletics	Strength and Resistance	Main part: Strength training on stairs and resistance games (example: Formula 1).	60 min
4	Athletics	Technics and Speed	Main part: Race modeling (pins at different distances). Speed: Short-duration speed.	60 min
	Swimming	Technics and Training	Main part: Learning basic aquatic motor skills (free games) and basic fundamentals of swimming.	60 min
	Athletics	Strength and Resistance	Main part: General physical condition circuit and resistance games (example: suicide).	60 min
5	Athletics	Technics and Speed	Main part: Long jump technics training. Speed: Short-duration speed (15 to 20 m).	60 min
	Handball	Technics and Training	Main part: Relationship with the ball (dribbling and feinting).	60 min
	Athletics	Strength and Endurance	Main part: General physical condition circuit and aerobic running (10–15 min).	60 min
6	Athletics	Technics and Speed	Main part: Block start training and race modeling. Speed: Various starts—sitting, lying, etc.	60 min
	M. Games	Technics and Training	Main part: Games with varied movements (stations: hopping, running, jumping, and throwing).	60 min
	Athletics	Strength and Endurance	Main part: Agility and coordination circuit and long jump and resistance games (relay).	60 min
7	Athletics	Technics and Speed	Main part: Relay training technics and circuit for agility and coordination (slalom).	60 min
	Handball	Technics and Training	Main part: Relationship with the ball (dribbling and feinting).	60 min
	Athletics	Strength and Endurance	Main part: Strength training using only body weight and aerobic running (10–15 min).	60 min
8	Athletics	Technics and Speed	Main part: High jump training. Speed: Short-duration speed (15 to 20 m).	60 min
	Gymnastics	Technics and Training	Main part: Basic fundamentals of gymnastics (improving jumping and rolling exercises).	60 min
	Athletics	Strength and Endurance	Main part: General physical condition circuit and resistance games (ex.: relay 4 × 400 m).	60 min
9	Athletics	Technics and Speed	Main part: Running technics (Skipping’s). Speed: Speed games (ex.: Divisions game).	60 min
	M. Games	Technics and Training	Main part: Object manipulation games and balance challenges with balls, hoops, and ropes.	60 min
	Athletics	Strength and Endurance	Main part: Multi-jumps in sandpit and aerobic running (10–15 min).	60 min
10	Athletics	Technics and Speed	Main part: Running technique training with cones and agility and coordination circuit.	60 min
	Swimming	Technics and Training	Main part: Learning basic aquatic motor skills (propulsion and manipulation exercises).	60 min
	Athletics	Strength and Endurance	Main part: Strength training using only body weight and aerobic running (15–20 min).	60 min
11	Athletics	Technics and Speed	Main part: High jump training. Speed: Short-duration speed (30 to 40 m).	60 min
	Handball	Technics and Training	Main part: Relationship with the ball and fundamental technical and tactical notions in game.	60 min
	Athletics	Strength and Endurance	Main part: General physical condition circuit and resistance games (ex.: Formula 1).	60 min
12	Athletics	Technics and Speed	Main part: Hurdles training technic and running at high speed with 4, 5, and 6 low hurdles.	60 min
	M. Games	Technics and Training	Main part: Circuit of games for cooperation, manipulation, locomotion, and coordination.	60 min
	Athletics	Strength and Endurance	Main part: Strength training using only body weight and aerobic running (15–20 min).	60 min

Note: M. Games—motor games.

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
