# Peer review of "Effects of a Twelve-Week Complementary Sports Program to Athletics Training on Motor Competence in Children Aged 6 to 10 Years Old—A Study Protocol"

_healthcare, 2025, doi:10.3390/healthcare13172111_

Round 1
Reviewer 1 Report
Comments and Suggestions for Authors
I read this manuscript and found it of certain interest. However, it needs revision:
1. The abstract mentions the use of Cohen's d effect size and Pearson's correlation coefficient. In contrast, the main text specifies a 2×2 repeated-measures ANOVA with η²p as the effect size measure and Bonferroni-adjusted post hoc tests. The abstract must be revised to align with the primary analysis plan described in the methods. Cohen's d may be appropriate for follow-up pairwise comparisons, but this should be clarified.
2. Furthermore, the sample size justification is absent. With n = 50 (25 per group), the study may be underpowered to detect moderate effect sizes, particularly given the expected variability in motor competence. From my calculations the least sample 60 with 30 in every group.
3. The sample of children enrolled in the Portuguese Athletics Federation athletics club introduces selection bias. It limits the generalizability to the broader pediatric population, particularly inactive children.
4. Although the training programs are detailed in Tables, there is no mention of how adherence to the protocol will be monitored.
5. The post-intervention assessment is scheduled 48 hours after the final session, which is appropriate to minimize acute fatigue effects. However, the manuscript notes assessments will occur between 18:30 and 19:30, while training for IG_B occurs at 17:30–18:30 on Thursdays. Care must be taken to avoid testing immediately after training.
6. There is an inconsistency in the complementary activities for IG_B - the methods section lists gymnastics, handball, swimming, and motor games, but the tables show variations without a clear rationale.
7. The statistical analysis plan mentions ANOVA but does not specify how it will handle multiple comparisons from the four KTK3+ subtests.
8. No mention of the randomization method, and no details about assessor blinding
9. Anthropometric protocols lack reliability measures.
10. The discussion prematurely assumes IG_B's superiority before data collection, which could indicate bias. The conclusion section also seems underdeveloped.
Reviewer 2 Report
Comments and Suggestions for Authors
Thank you for submitting this to Healthcare. This study aimed to evaluate the effects of a 12-week exercise training program and a multi-activity training program on motor skills in children aged 6 to 10 years.
The author submitted this study as a protocol study. The author stated that the results and conclusions will be published later, but in my opinion, it is appropriate to submit this study as original research including the results and conclusions. In particular, the author conducted the experiment for 12 weeks.
In order to submit it as a protocol study, the author must have developed a new protocol, and the protocol must be clearly different from existing ones, and the theoretical principles must be scientific. Therefore, the protocol development process must be recorded in detail.
Looking at Tables 1 and 2 written by the author, it is difficult to judge that this study is a unique protocol.
The design of this study differs in the inclusion of sports such as swimming and handball. I do not think there is a ‘better exercise’ or a ‘lesser exercise’ for exercise performed on children aged 6 to 10. Of course, there will be motor functions that improve more when exercising sports. However, if the results are in the form of ‘this is a better exercise’ for children, this will create another issue.
However, if this design is maintained, the author needs to change the title. Since the purpose is to compare Table 1 and Table 2 exercises, a title that can express this nature is required.
Introduction: The following contents should be included. The growth characteristics, motor function development characteristics, and physiological characteristics that appear at ages 6-10 should be described. And the results of other studies that studied the effects of exercise according to these characteristics should be specifically cited. In particular, since the author selected gymnastics, handball, swimming, and motor games as intervention sports, the previous literature should be referenced to see how these sports affect motor function in children aged 6-10.
Research Methods: The current research methods are described in relatively detail. To improve the reader’s understanding, it is recommended to use the same font size and number the sections, such as 2.1, 2.2, etc.
Results: This section must be included for the author’s research to be published.
Discussion: The difference between Introduction and Discussion is as follows. If Introduction is about ‘Why is this study necessary?’, Discussion requires interpretation of ‘Why did the results come out like this?’ Therefore, Discussion should be based on the developmental and physiological characteristics mentioned in Introduction and scientifically written based on evidence. And there should be a comparison of the results with other similar studies.
Please show the process and results of calculating the sample size.
The citation and reference style deviate from MDPI format. Authors are requested to check this part again. And please use references that are within 5 or 10 years.
The sentence expression should be changed to a more scientific and academic style.
Reviewer 3 Report
Comments and Suggestions for Authors
This study is a well-designed protocol that aims to examine the effects of a 12-week sports program on motor competence (MC) in children aged 6-10 years. The study aims to contribute to the literature by comparing two different intervention groups based on athletics and multi-activity. However, it requires major revision due to some methodological and reporting shortcomings.
Purpose and Importance of the Study: The role of MC in child development and the importance of early interventions were clearly defined. A randomized controlled experimental design, age-appropriate measurement tools (KTK3+) and detailed intervention protocols (IG_A and IG_B) were used. There was strong agreement with theories of motor development (e.g., Stodden and Clark models) and the literature.
- Sample Size and Power Analysis: How the sample size of 50 participants was determined and the adequacy of the statistical power were not explained. Recommendation: Power analysis with a tool such as G*Power should be added.
- Although the KTK3+ test used provides normative values by age and sex, the sensitivity of these values may be reduced over a wide age range. Analyses should include analysis of covariance (ANCOVA) for age and sex or comparisons in subgroups.
- Details of the randomization procedure (e.g., block randomization) and blinding (assessor blinding) are missing. Recommendation: The CONSORT flowchart and randomization steps should be clarified.
- Factors such as diet, sleep patterns or out-of-school physical activity were not controlled. Recommendation: These variables should be measured by questionnaires and included in the analysis.
Multiple measurements with additional tests such as TGMD-2 or MABC would have improved the quality of the study. Inter-rater reliability and pre-test training process were not specified for ICC3+. Practitioner training and ICC analyses should be reported.
- “Not applicable”, but the ethics committee approval number and date should be added since the study was conducted with human participants.
- Instead of “not applicable”, it should be explained how the data will be accessible (e.g., anonymized dataset upon request).
Round 2
Reviewer 1 Report
Comments and Suggestions for Authors
The authors responded to all issues
Reviewer 2 Report
Comments and Suggestions for Authors
I do not have any comments.8
Reviewer 3 Report
Comments and Suggestions for Authors
Thanks for your efforts